# Resilience in Infrastructure Systems: A Comprehensive Review

**Wenque Liu** [1], **Ming Shan** [1,*], **Sheng Zhang** [1], **Xianbo Zhao** [2] and **Zhao Zhai** [3]

[1] School of Civil Engineering, Central South University, 68 South Shaoshan Road, Changsha 410004, China; 1202150227@csu.edu.cn (W.L.); zhsh1230@126.com (S.Z.)

[2] School of Engineering and Technology, Central Queensland University, 400 Kent St., Sydney, NSW 2000, Australia; b.zhao@cqu.edu.au

[3] School of Traffic and Transportation Engineering, Changsha University of Science and Technology, 960 Wanjiali Road, Changsha 410205, China; zhao.zhai@csust.edu.cn

* Correspondence: ming.shan@csu.edu.cn

**Abstract:** When encountering crisis events, systems, organizations, and people must react and handle these unpredictable events. Under these circumstances, important social functions and infrastructures must be restored or adapted as quickly as possible. This capacity refers to resilience. Although considerable research has been conducted on the resilience of infrastructure systems over the past years, a critical review of these studies remains lacking. Therefore, this study aims to bridge the knowledge gap by presenting a comprehensive review of infrastructure research conducted in the past decade, namely, from 2011 to 2021. On the basis of a systematic search, this study identified 222 journal articles investigating infrastructure resilience. A review of the identified papers revealed five research streams in the area of infrastructure resilience (IR), namely, the assessment of infrastructure resilience, improvement of infrastructure resilience, conceptualizing infrastructure resilience from various perspectives, factors influencing infrastructure resilience, and the prediction of infrastructure resilience. This study also presented some directions that future research can pursue. These directions include analyzing factors influencing infrastructure resilience based on simulation, assessing the resilience of green infrastructure, improving the resilience of interdependent infrastructure, and predicting the resilience of infrastructure based on empirical research.

**Keywords:** infrastructure resilience; resilience improvement; simulation; green infrastructure

## 1. Introduction

Generally, infrastructure refers to the fundamental facilities and systems in urban and rural areas, such as railways, roads, tunnels, bridges, power grids, energy, electricity, telecommunications, water supplies, sewers, and organization [1]. They are used for managing and controlling environmental systems to ensure that critical services and resources are available when and where they are required [2]. For example, water infrastructure, such as dams, drainage culverts, reservoirs, and pumps, are designed to mitigate the adverse effects of extreme rainfall, storm surges, or floods [3,4]. Power infrastructure plays a pivotal role in maintaining the operation of other critical infrastructure systems, including water, transportation, and telecommunications [5]. As a system responsible for the service of countries, cities, and other regions, infrastructure contributes to maintaining people's basic needs and meeting their daily requirements, and it also plays an important role in responding to various disasters [6].

In recent years, urban and rural areas around the world have suffered from different types of disasters [7]. Based on the report of the United Nations International Strategy for Disaster Reduction [8], natural disasters, such as earthquakes, floods, blizzards, and hurricanes, have caused economic losses of over USD 300 billion annually worldwide. These natural disasters are a major source of vulnerability to urban assets [9]. For instance, the super hurricane "Sandy" in New York in 2012 caused severe damage to local infrastructure,

and residents suffered a great economic loss of over USD 70 billion from this disaster [10]. In 2017, the flood caused by Hurricane "Harvey" in Texas and Louisiana destroyed many transportation infrastructure systems, forcing a great number of people to evacuate from their homes, with an economic loss of over USD 190 billion [11]. Such catastrophic events attract growing attention to enhancing the resilience of infrastructure systems.

The concept of resilience was first used in mechanics, and it refers to the ability of an object to return to its original state when it is subjected to an external force to produce a shape change without breaking [12–14]. Subsequently, it has been widely adopted in multiple fields such as ecology, psychology, sociology, and public management [15–17]. Generally, resilience is usually defined as the ability of a system, society, or region to react, absorb, adapt to, and recover from disruptive events rapidly and effectively [18]. Infrastructure resilience (IR) is the ability of system-wide recovery after disasters when infrastructure systems suffer local damage after extreme natural or human-made disruptive events [19]. Compared with other systems, infrastructure systems are more interdependent, and their interdependence can affect their resilience to routine disruptions and extreme events [20]. As asserted by Shakou et al. [21], if infrastructure systems lack resilience, the entire city or region will be extremely vulnerable. Thus, maintaining and enhancing the resilience of infrastructure systems is critical, especially when people suffer from tremendous disruptions.

Great attention has been paid to the field of infrastructure resilience, and various types of studies related to this field have also emerged, including a few review papers. However, these reviews primarily focus on a specific category of infrastructure or the research methods used in studies of infrastructure resilience [22–24], and a comprehensive review of IR-related studies is relatively lacking. This study aims to bridge the knowledge gap by conducting a comprehensive literature review of the existing studies in the field of IR, hence summarizing the major research streams in the area and proposing possible directions for future research. This study contributes to the current body of knowledge by reviewing state of the art of IR and revealing the potential areas for future research. In addition, this study is useful to the practice as well, as it includes the latest approaches that can be used to build, maintain, and improve IR in practice.

## 2. Research Methodology

Over the past decade (2011–2021), considerable research efforts have been conducted to investigate IR. According to Ke et al. [25], research teams in the area of civil engineering and management normally submit their best research findings to peer-review journals with a good reputation for publication. Therefore, this study chose to search and then review the research papers published in highly ranked journals to present state of the art research on infrastructure resilience.

This study adopted a four-phase search strategy derived from a structured method adopted by Ke et al. [25] and Hu et al. [26] to identify and analyze the journal papers valid for review and the framework of the literature review is shown in Figure 1. In phase one, an exploratory search was conducted on the website of Web of Science and Scopus. These databases were selected because they include more than 1000 peer-review journals that are highly recognized by researchers worldwide [26]. To obtain the maximum number of papers on IR, a concise code *infrastructure resilience* was searched in the *Title* field on the website of Web of Science and Scopus in the timespan from 2011 to 2021; then, the findings of the enquiry of Scopus and Web of Science were cross-checked. Lastly, 510 papers were obtained in the search of Phase 1.

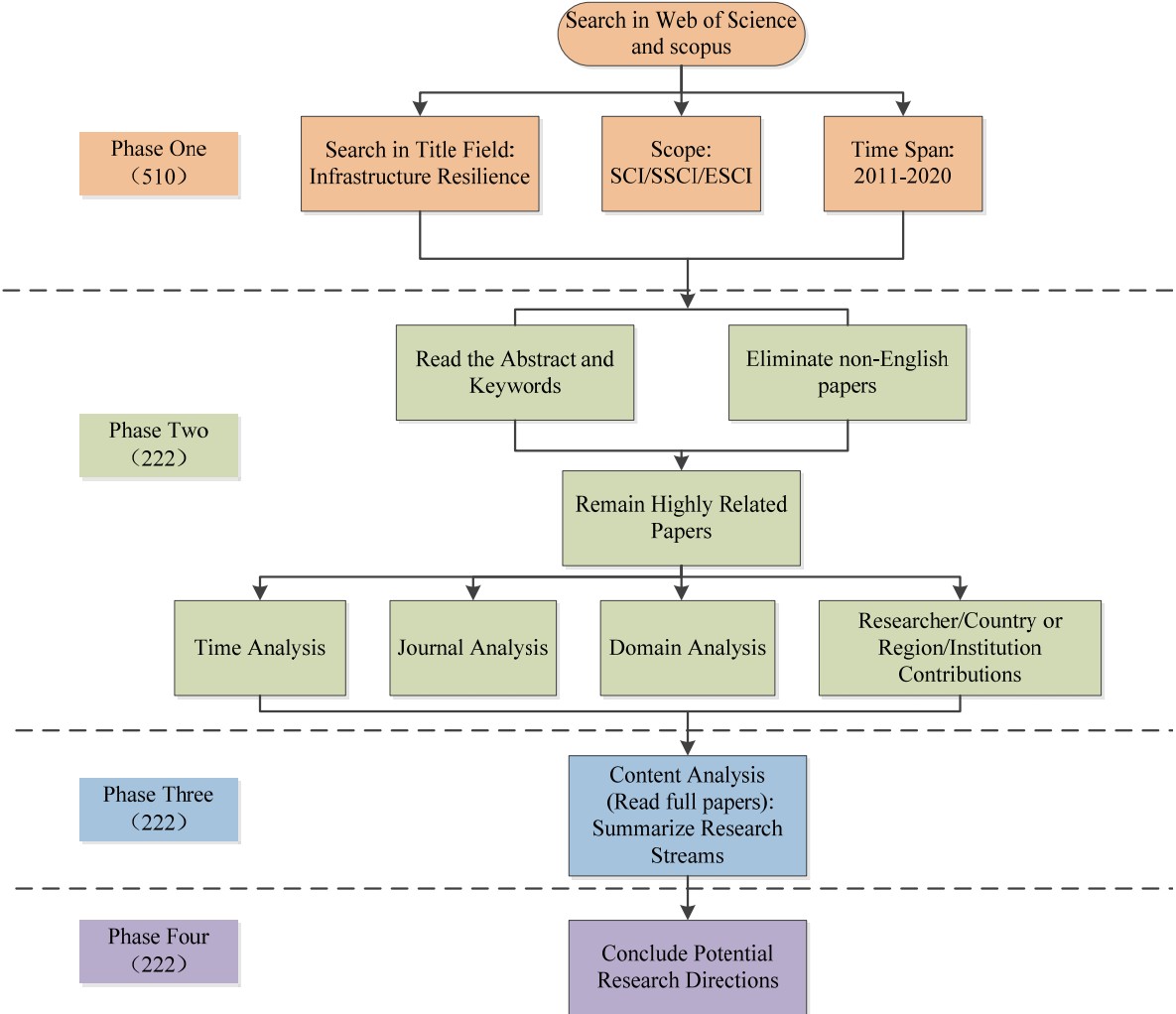

**Figure 1.** Literature review framework.

In phase two, a thorough review of the abstracts and titles and keywords of the 510 papers was conducted. Papers that were highly related to IR and written in English were retained, and those with weak or no correlation with IR and written in languages other than English were eliminated, resulting in a remaining 222 papers. The 222 papers were quantitatively analyzed to determine their contributions by time, journal, domain, country/region, institution, and researcher. Research contributions of a country (region) and its affiliated institutions and researchers can yield a collective view of the current status of a certain research field [27]. Quantifying the research contributions of researchers is the basis for analyzing the contributions of institutions and countries (regions). When checking the contributions of researchers, the scoring method developed by Howard, Cole, and Maxwell [28] was used for those studies having multiple authors. In this method, the credit of the authors listed in the same paper is calculated based on the order of authorship, as shown in Formula (1):

$$\text{Score} = \frac{1.5^{n-i}}{\sum_{i=1}^{n} 1.5^{n-i}} \tag{1}$$

where *n* is the number of authors of the paper and *i* is the order of the specific author. Given that each paper has a score of one point, the detailed score matrix generated by the method is provided in Table 1.

**Table 1.** Score matrix for papers with multiple authors.

| Number of Authors | Order of Specific Authors | | | | |
|---|---|---|---|---|---|
| | **1** | **2** | **3** | **4** | **5** |
| 1 | 1.00 | N/A | N/A | N/A | N/A |
| 2 | 0.60 | 0.40 | N/A | N/A | N/A |
| 3 | 0.47 | 0.32 | 0.21 | N/A | N/A |
| 4 | 0.42 | 0.28 | 0.18 | 0.12 | N/A |
| 5 | 0.38 | 0.26 | 0.17 | 0.11 | 0.08 |

In addition to assessing the contributions of researchers, institutions, and the relevant contributing countries or regions, major research streams of the existing infrastructure resilience literature and possible directions for future research in the field were summarized and presented in phase three and phase four as well. Although these analyses are unable to cover all details of the 222 papers, they can build up the overall picture of the infrastructure resilience studies from 2011 to 2021, and they will be useful both to academia and in practice.

## 3. Overview of Infrastructure Resilience Research: Distribution Analysis

### 3.1. Publications over Time

By plotting the number of publications per year for the identified papers, a clear trend can be seen, as shown in Figure 2. The plot shows that there were relatively few publications per year from 2011 until 2014. From 2015 onward, especially in 2020, a significant increase in the number of publications on infrastructure resilience occurs. One probable reason for the increase in research interest is the large mass of disasters mentioned at the beginning of the paper, which has strongly attracted the attention of the global research community.

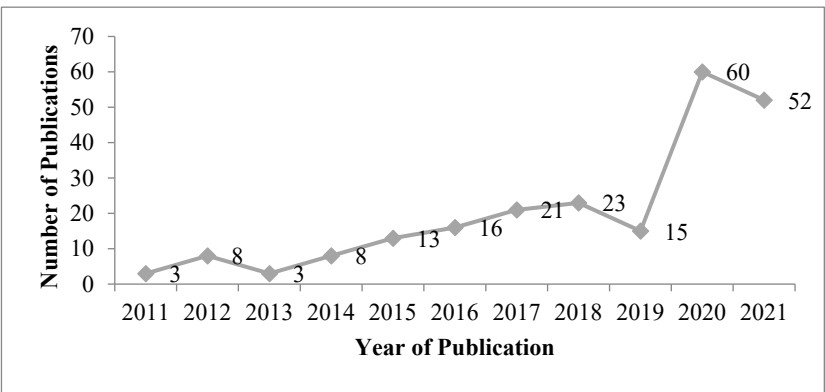

**Figure 2.** Number of publications over time.

### 3.2. Publications by Journal

The identified papers were published in 102 journals, suggesting a broad range of peer-review journals have published infrastructure resilience studies in the past decade. However, some journals published in this area more intensively. Table 2 present the top 10 journals that published the most infrastructure resilience papers. Among these journals, *Reliability Engineering and Sustainability* took the lead and was followed by the *Journal of Infrastructure Systems, Safety Science, International Journal of Disaster Risk Reduction, Risk Analysis, Sustainable and Resilient Infrastructure, IEEE Systems Journal, Computer-aided Civil and Infrastructure Engineering,* and *Natural Hazards*. All these 10 journals published 62 papers in the aggregate, accounting for 36.47% of the total papers obtained from the literature search.

**Table 2.** Top 10 journals publishing infrastructure resilience papers.

| No. | Journal Title | Numbers of Papers | % |
|-----|---------------|-------------------|---|
| 1 | Reliability Engineering and System Safety | 14 | 8.24 |
| 2 | Sustainability | 8 | 4.71 |
| 3 | Journal of Infrastructure Systems | 6 | 3.53 |
| 4 | Safety Science | 6 | 3.53 |
| 5 | International Journal of Disaster Risk Reduction | 6 | 3.53 |
| 6 | Risk Analysis | 5 | 2.94 |
| 7 | Sustainable and Resilient Infrastructure | 5 | 2.94 |
| 8 | IEEE Systems Journal | 4 | 2.35 |
| 9 | Computer-aided Civil and Infrastructure Engineering | 4 | 2.35 |
| 10 | Natural Hazards | 4 | 2.35 |
| Total | | 62 | 36.47 |

*3.3. Publications by Infrastructure Type*

As shown in Figure 3, the identified papers can be separated into two groups according to the types of infrastructures investigated. One group contains 81 papers (47.65% of the total) investigating the resilience problem in critical, general, and interdependent infrastructures. The other group contains 89 papers (52.35% of the total) investigating resilience in a specific type of infrastructure such as water, electrical and power, transport and logistics, health, education, urban, civil and lifeline, transportation, highway, maritime, energy, communication, bridge, mountain, organization, and railway infrastructure. Particularly, it can be seen from Figure 3 that urban and green, water, electrical and power, transportation, civil and lifeline, transport and logistics, marine, highway and road, and energy infrastructures are all investigated by at least four papers, suggesting the resilience issue in these areas are very critical and has high topicality.

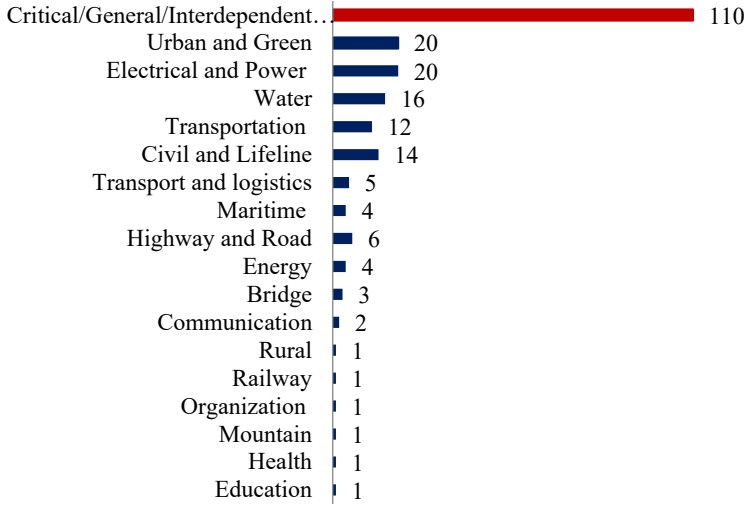

**Figure 3.** Number of publications by infrastructure type (222 in total).

*3.4. Top Contributing Researchers/Institutions/Countries or Regions*

The contributions of the researchers to a given topic can be measured by calculating their cumulative scores based on the score matrix presented in Table 1. For example, if a researcher has the first authorship in one paper of three authors and the second authorship in another paper of four authors, then this researcher can obtain a score of 0.75 (0.47 + 0.28) on the basis of Table 1. If the number of authors of a paper is more than five, then the scores of every author of this paper can be directly calculated by Equation (1). According to the calculating rules, the scores of the researcher were calculated, and the top 10 researchers involved in research on infrastructure resilience are listed in Table 3. As shown in Table 3, Min Ouyang from Huazhong University of Science and Technology received the highest

score of 3.22 by contributing four publications; Ali Mostafavi from Texas A&M University received the second-highest score of 1.89 by contributing five publications, and Sara Meerow gained the third-highest score of 1.60 by contributing two publications. In addition, among the top 10 researchers, two were from China, four were from the United States, two were from Italy, one was from Canada, and one was from England. It can be seen that most of the top researchers are from developed countries where critical infrastructure systems are already established.

**Table 3.** Top 10 researchers involved in the research of IR.

| Score | Ranking | Researchers | Number of Papers | Research Institutions | Country/Region |
|-------|---------|-------------|------------------|-----------------------|----------------|
| 3.22 | 1 | Min Ouyang | 4 | Huazhong University of Science and Technology | China |
| 1.89 | 2 | Ali Mostafavi | 5 | Texas A&M University | United States |
| 1.60 | 3 | Sara Meerow | 2 | Arizona State University | United States |
| 1.07 | 4 | Kong Jingjing | 3 | Western University | Canada |
| 0.79 | 5 | X Liu | 2 | Shanghai Jiao Tong University | China |
| 0.77 | 6 | Hadi Nazarnia | 2 | University of Florida | United States |
| 0.72 | 7 | Leonardo Duenas-Osorio | 2 | Rice University | United States |
| 0.68 | 8 | Paolo Trucco | 2 | Polytechnic University of Milan | Italy |
| 0.66 | 9 | Pierluigi Mancarella | 2 | University of Manchester | England |
| 0.64 | 10 | Boris Petrenj | 2 | Polytechnic University of Milan | Italy |

As mentioned above, by using the method of calculating researchers' scores, the score of an institution can also be calculated based on the scores of researchers who belong to this institution. For example, if two researchers from the same institution contribute to one paper, with one researcher attaining a score of 1 and the other attaining a score of 2, then the institution obtains a score of 3 (1 + 2). Table 4 presents the research institutions attaining the top 10 scores. As shown in Table 4, the University of Florida ranked first among all identified research institutions with a score of 4.67, followed by Arizona State University, Texas A&M University, Huazhong University of Science and Technology, University of Pittsburgh, Rice University, Polytechnic University of Milan, University of Manchester, Naval Postgraduate School, and Western University.

**Table 4.** Top 10 research institutions publishing papers on IR.

| Score | Ranking | Research Institutions | Country/Region | Number of Papers | Researchers |
|-------|---------|-----------------------|----------------|------------------|-------------|
| 4.67 | 1 | University of Florida | United States | 3 | 9 |
| 3.80 | 2 | Arizona State University | United States | 3 | 10 |
| 3.58 | 3 | Texas A&M University | United States | 3 | 7 |
| 3.22 | 4 | Huazhong University of Science and Technology | China | 4 | 1 |
| 3.00 | 5 | University of Pittsburgh | United States | 2 | 5 |
| 1.33 | 6 | Rice University | United States | 2 | 3 |
| 1.32 | 7 | Polytechnic University of Milan | Italy | 2 | 2 |
| 1.25 | 8 | University of Manchester | England | 2 | 2 |
| 1.17 | 9 | Naval Postgraduate School | United States | 2 | 3 |
| 1.07 | 10 | Western University | Canada | 2 | 1 |

According to Hong et al. [27], the number of academic research publications in a country or region reflects the degree of industrial development and practice progress in the research field of the region. Therefore, an analysis of the research contributions of a country or region was conducted in this review. Similar to the calculation of the scores of research institutions, the scores of every country or region were calculated. Table 5 present the scores of the contributing countries or regions, together with the number of papers and the number of the researchers involved. It could be seen from Table 5 that the United States received the highest score (i.e., 45.04), suggesting the United States contributes most to the

research of infrastructure resilience. Moreover, as shown in Table 5, the score of the United States (i.e., 45.04) is significantly higher than that of the United Kingdom, ranking second, suggesting research in the United States is far ahead of those in the remaining countries.

**Table 5.** The research contribution of country/region in papers of IR.

| Score | Ranking | Country/Region | Number of Papers | Researchers Involved |
|---|---|---|---|---|
| 45.04 | 1 | The United States | 50 | 103 |
| 10.73 | 2 | The United Kingdom | 7 | 10 |
| 9.06 | 3 | China | 10 | 16 |
| 6.74 | 4 | Italy | 8 | 13 |
| 5.65 | 5 | Canada | 6 | 11 |
| 4.00 | 6 | Switzerland | 6 | 7 |
| 3.19 | 7 | Australia | 3 | 9 |
| 2.68 | 8 | Netherlands | 2 | 8 |
| 2.21 | 9 | France | 3 | 8 |
| 2.00 | 10 | Spain | 2 | 3 |
| 1.47 | 11 | New Zealand | 2 | 5 |
| 1.39 | 12 | Iran | 2 | 5 |
| 1.36 | 13 | Sweden | 2 | 4 |
| 1.07 | 14 | Denmark | 2 | 2 |
| 1.00 | 15 | Nigeria | 1 | 1 |
| 1.00 | 16 | Latvia | 1 | 1 |
| 1.00 | 17 | Norway | 1 | 2 |
| 0.95 | 18 | Brazil | 1 | 3 |
| 0.79 | 19 | Mauritius | 1 | 2 |
| 0.72 | 20 | Belgium | 1 | 3 |
| 0.60 | 21 | Japan | 1 | 1 |
| 0.51 | 22 | Germany | 1 | 3 |

## 4. Infrastructure Resilience Research: Major Research Streams

After conducting an overview of the 222 papers, content analysis was used as a means for determining the primary streams of this IR-related literature. The statements, data, figures, and tables existing in these studies were analyzed in detail during this process. Finally, five streams of the infrastructure resilience research were identified, as shown in Table 6. These research streams include the assessment of infrastructure resilience, the improvement of infrastructure resilience, conceptualizing infrastructure resilience from various perspectives, factors influencing infrastructure resilience, and the prediction of infrastructure resilience. Table 6 reflects the number of studies on each stream in the field of IR. Among 222 identified pieces of literature, over half of the total focused on the assessment of IR, with a proportion of 54.1%. The number of research on improving IR occupies the second position, taking up 27.0%. Literature on decomposing IR and Influencing IR are, respectively, responsible for 10.4% and 6.3%. Additionally, the number of research on predicting IR is the least, only accounting for 2.2%. The major research efforts carried out in these streams are elaborated on in the following sections.

**Table 6.** Major research streams of infrastructure resilience research.

| Research Streams | Assessment of Infrastructure Resilience | Improvement of Infrastructure Resilience | Conceptualizing Infrastructure Resilience from Various Perspectives | Factors Influencing Infrastructure Resilience | Prediction of Infrastructure Resilience | Total |
|---|---|---|---|---|---|---|
| 2011 | 1 | 1 | 1 | 0 | 0 | 3 |
| 2012 | 5 | 2 | 1 | 0 | 0 | 8 |
| 2013 | 3 | 0 | 0 | 0 | 0 | 3 |
| 2014 | 7 | 1 | 0 | 0 | 0 | 8 |
| 2015 | 9 | 3 | 1 | 0 | 0 | 13 |
| 2016 | 8 | 4 | 3 | 1 | 0 | 16 |
| 2017 | 10 | 7 | 2 | 0 | 2 | 21 |
| 2018 | 11 | 6 | 2 | 3 | 1 | 23 |
| 2019 | 8 | 6 | 1 | 0 | 0 | 15 |
| 2020 | 30 | 15 | 11 | 4 | 0 | 60 |
| 2021 | 28 | 15 | 1 | 6 | 2 | 52 |
| Total | 120 | 60 | 23 | 14 | 5 | 222 |
| % | 54.1 | 27.0 | 10.4 | 6.3 | 2.2 | 100 |

*4.1. Assessment of Infrastructure Resilience*

Assessing infrastructure resilience is critical to determining preventive measures to mitigate the consequences caused by various disruptive events [29]. A large number of researchers have assessed IR responding to disruptions by adopting different analytical tools. For example, Levenberg et al. [30] quantified and assessed the resilience of networked pavement infrastructure by modeling a set of possible network performance scenarios in a destructive meteorological scenario with a known probability of occurrence; each scenario was defined according to the severity and type of damage (climate or geology, operation, natural deterioration, and terrorism), as well as current weather conditions, temperature, precipitation, and visibility conditions. Sun, Stojadinovic, and Sansavini [31] put forward an agent-based modeling framework for the seismic resilience assessment of integrated civil infrastructure systems under the scenario of an earthquake. Barabadi et al. [32] assessed the resilience of health infrastructure before and after COVID-19. Xu, Cong, and Proverbs [33] proposed a multidimensional evaluation index system for assessing the resilient capacity of infrastructure systems to cope with the extreme weather in Wuhan.

Quantifying resilience is a challenging problem [34]. Bruneau et al. [16] proposed a comprehensive assessment framework of resilience, pointing out that 11 aspects need to be considered when assessing resilience (Figure 4), including 4 dimensions (i.e., technical, organizational, social, and economic), 4 basic properties (i.e., robustness, rapidity, redundancy, and resourcefulness), as well as 3 outcomes (i.e., more reliable, faster recovery, lower consequences). Huck, Monstadt, and Driessen [35] applied the 4R concept integrating robustness, rapidity, redundancy, and resourcefulness in assessing the resilience of power infrastructure in order to establish disaster management. Toroghi and Thomas [36] used a framework including five metrics (robustness, resourcefulness, redundancy, rapidity, and readjust-ability), which assesses the resilience of electric infrastructure systems using the 4R concept and the readjust-ability.

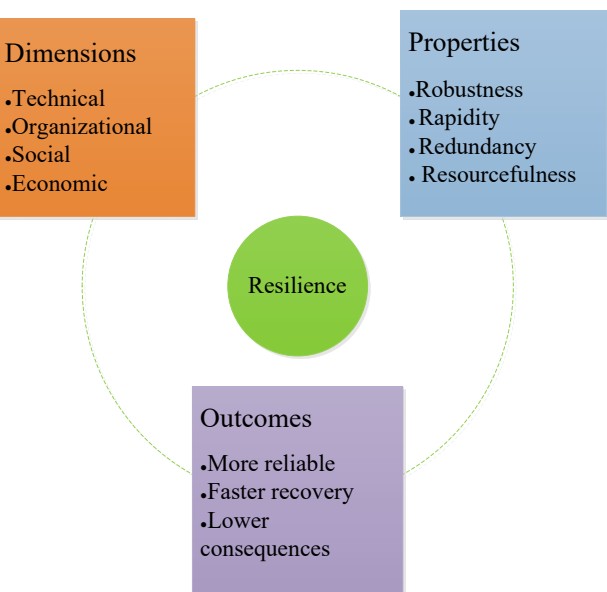

**Figure 4.** Assessment framework of resilience (11 aspects).

Robustness is one of the most important properties of resilience, referring to the ability of the system to survive under abnormal and hazardous conditions [37]. When one or more of the capacities of the interconnected system are exhausted, a resilience failure event will occur, which firstly relates to the system's robustness [38]. A variety of researchers assessed the resilience of infrastructure in terms of robustness [39–41]. Homayounfar et al. [40] used a stylized dynamical model for assessing the connection between resilience and robustness of coupled infrastructure systems, stating that robustness is a component of resilience; this

model simulates how the robustness and resilience of critical infrastructure systems respond to shocks in state variables change with parameters. Huizar et al. [41] used robustness as a metric for quantifying the resilience of supply and demand of water infrastructure systems, providing a reference for water infrastructure planning and design.

Redundancy can enhance the reliability of a system through the duplication of its critical components or functions, which is one of the metrics measuring the system's resilience [42]. Perz et al. [43] adopted a three-country border in the southwestern Amazon that was integrated by a highway as a case study, and they used rural survey data to assess the relationship between the connectivity of community and town and socioecological resilience, exploring the effects of resilience and connectivity on infrastructure. Capacci and Biondini [44] proposed a probabilistic framework for assessing the life-cycle seismic resilience of infrastructure networks integrating the aging bridges and transportation roads, thus exploring the approach to strengthening the redundancy of infrastructure networks.

Rapidity refers to the capacity of a system to meet priorities and achieve goals in time after disruptions to control enormous losses, which indicates minimizing the time required to recover to full system operations and productivity [45]. A number of researchers evaluated IR using this property [46–49]. Cimellaro et al. [46] presented recovery curves based on the time series recorded from March 11 to April 26 during the 2011 Tohoku Earthquake in Japan to assess the resilience of physical infrastructure; the recovery curves, respectively, indicate the restoration ratio between the number of households without service and the total number of households in terms of three categories of lifelines along with the time series, including power delivery, water supply, and city gas delivery. Argyroudis et al. [48] adopted a multi-hazard assessment framework to quantify the resilience of critical infrastructure systems; considering the impacts of cascading hazards, where the subsequent hazard is triggered by the initial hazard simultaneously or within a short period, and the restoration commences after the completion of the multiple hazard sequences, this framework measures the vulnerability of the assets to risks and the rapidity of the restoration in the scenario of cascading hazards.

As a property of measuring resilience, resourcefulness is the ability of a system to allocate resources rationally to minimize the impacts of hazards and improve its performance [50]. Vadali et al. [51] proposed the optimal approach to maintaining infrastructure resilience by using a bi-national dynamic traffic assignment model; this model provides a timely insight into the daily travel impact and economic cost of an unexpected disruption to the ports-of-entry infrastructure. Lau et al. [52] identified the most efficient approach to meeting the demand shortage of critical infrastructure and maintaining the resilience of the power grid by establishing a grid optimization model, which consisted of the low-level (micro-grid) and mid-level voltage grid components in urban power grids for disaster recovery.

Compared with the research assessing IR from a perspective of a single property, the majority of reviewed papers developed and quantified the resilience of infrastructure systems based on comprehensive properties. For example, Shafieezadeh and Burden [53] proposed a probabilistic framework for the scenario-based resilience assessment of infrastructure systems; this method considered the uncertainty in the process, including the correlation of seismic intensity measurement, vulnerability assessment of structural components, estimation of maintenance demand, maintenance process, and service demand. Zhu et al. [54] used a flexible assessment framework that comprised eight metrics adapted from existing research: vulnerability, expectations, redundancy, adaptability, rapidity, intelligence, cross-scale interaction, and learning culture, to assess the resilience of the power and water infrastructure systems in Kathmandu Valley during the 2015 Gurkha earthquake.

### 4.2. Improvement of Infrastructure Resilience

A resilient infrastructure system is supposed to minimize the probability of failure, possess the redundant connectivity, shrink the recovery time, and limit impact propagation, which corresponds to four properties including robustness, redundancy, rapidity, and

resourcefulness (the relationship between a resilient infrastructure and four properties of resilience is shown in Figure 5) [11]. A total of 45 papers investigated how the resilience of infrastructure systems can be improved and what strategies can be used to improve infrastructure resilience based on the four aspects. The majority of papers conducted the research on improving the comprehensive resilience of infrastructure resilience.

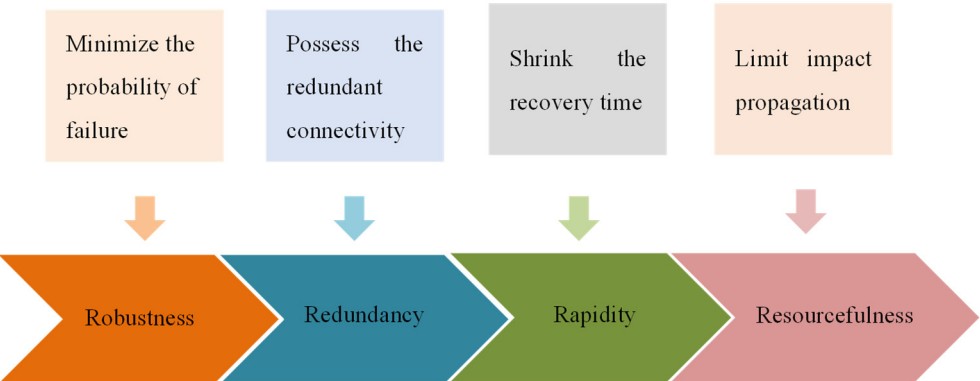

**Figure 5.** The relationship between resilient infrastructure and four properties of resilience.

In terms of minimizing the probability of failure, the infrastructure systems are supposed to enhance the robustness and stability of responding to hazards or attacks. Several studies provided strategies for improving the resilience to resist various hazards using different methods [55,56]. Zhao et al. [55] designed non-homogeneous hidden Markov models for resilience metrics in different damage scenarios such as natural disasters, man-made accidents, or violent attacks, aiming to propose an optimized scheduling strategy to improve and maximize the resilience of dynamic infrastructure systems. Johansen and Tien [56] adopted the Bayesian Network methods for probabilistic modeling of interdependencies, which could probabilistically infer which interdependencies have the most critical impacts and prioritize components for repair or reinforcement, facilitating and providing strategies for improving the resilience of infrastructure systems when encountering natural attacks and targeted attacks.

There are previous studies dedicated to enhancing urban robustness using green infrastructure systems [57,58]. Staddon et al. [57] stated that green infrastructure systems make great contributions to improving urban resilience; they conducted a high-level global overview of the contributions of green infrastructure to urban resilience and illustrated the challenges of reinforcing this green infrastructure-based pathway for seeking strategies of enhancing the urban robustness. Lee et al. [58] simulated the impact of reduced stormwater runoff in the Gangnam district of Seoul through the application of green infrastructure and proposed a green infrastructure-based strategy on the basis of the flood-adaptive design strategy and simulation results for enhancing the resilience of urban infrastructure systems to the flood.

The interdependence and connectivity of systems are mutually beneficial under normal operating conditions, which also enhance the ability of urban infrastructure networks to resist extreme events, such as terrorist attacks and natural disasters [59]. According to graph theory, connectivity is defined as the minimum number of elements (nodes or edges) that need to be removed in order to separate the remaining nodes into isolated subgraphs [60]. Generally, a greater number of interconnecting paths between two nodes can contribute to the lower isolation and higher accessibility of the infrastructure system, as well as the greater redundancy of the infrastructure network [61]. The performance of the infrastructure network can be improved as network connectivity grows, such as by increasing redundancy and adding the capacity of important links and interconnected nodes [62].

As an extremely resilient system, it will recover rapidly after disruptions. Freeman and Hancock [63] proposed distributed smart renewable energy micro-grid systems that mitigated adverse impacts by outage prevention and rapid service restoration, enhancing

the resilience of energy and communication infrastructure in rural and regional Australia. Yang et al. [64] proposed a multi-mode restoration model for enhancing the resilience of critical infrastructure systems to achieve the target of optimum post-disruption restoration under uncertainty, which included four steps: setting the boundaries of resilience analysis through prescribed standards; establishing the life cycle of infrastructure resilience management; defining physical-based infrastructure system functional modeling; designing an interface between interdependent infrastructure systems.

During the recovery period of disrupted infrastructure, rational resource allocation can minimize the economic and social impacts. Zhang, Kong, and Simonovic [65] firstly proposed an optimal allocation model of infrastructure recovery resources, assisting decision-makers in understanding the effects of resources allocation better and to decide on the adoption of allocation strategies after a disruptive event. Kong, Zhang, and Simonovic [66] then proposed a two-stage restoration resource allocation model to develop the optimal strategies for enhancing the resilience of interdependent infrastructure systems, with two goals about quickly restoring the dynamic resilience of infrastructure systems to meet the basic requirements of users in the first stage and minimizing the total loss in the subsequent recovery process in the second stage.

### 4.3. Conceptualizing Infrastructure Resilience from Various Perspectives

The word resilience originates from the Latin word "resiliere", which means to "bounce back" [67]. The concept of resilience is applied in distinct fields with different definitions and explanations. In the field of infrastructure, the knowledge body referring to resilience was explored and excavated in recent years. Twenty-three identified papers focused on the relevant concepts of IR from a variety of perspectives. For example, Alderson, Brown, and Carlyle [68] introduced the term "operational resilience" to define resilience as the ability that a system adapts its behaviors to maintain the continuity of function (or operations) when disruptive events occur. Mostafavi and Inman [69] used a comprehensive survey of the state transportation agencies in the USA to explore the concept of operational resilience of transportation infrastructures; they finally identified the major features of operational resilience of transportation infrastructures, including the availability of funding, integration of efforts across different units in the organization, use of risk and vulnerability approaches, as well as the adoption of resilience indices for decision making. Rogers et al. [70] concluded the concepts of the resilience of ecology/ecosystem, economy, physical infrastructure/engineering, community/society, and government systems for providing evidence to build the resilience of various infrastructures; the concepts that he summarized are listed in Table 7 to decompose local infrastructure resilience from distinguished perspectives.

**Table 7.** Concepts of local infrastructure resilience from different perspectives.

| Concept | Perspective |
|---|---|
| The resilience of an ecosystem is a measure of the durability of a system and of its ability to absorb changes and disturbances and still maintain the same relationships between population or state variables. | ecology/ecosystem |
| The resilience of the economy must owe to two characteristics: "intrinsic" and "adaptive"; the former includes responding strategies and advantages under normal circumstances, while the latter refers to the ability to take innovative actions under unpredictable circumstances. | economy |
| The resilience of engineering can make it flexible to cope with different disasters. | physical infrastructure/engineering |
| The resilience of the community is related to humans, manifesting itself in meetings and dialogue, communication and training, skills and information, and action when disaster strikes. | community/society |
| The resilience of government is usually explained as encompassing actions to prevent, protect against, and prepare for natural and man-made disasters. | government |

### 4.4. Factors Influencing Infrastructure Resilience

Infrastructure systems constantly face the impacts of a variety of factors, which undoubtedly brings many challenges to IR [71]. A few researchers have worked on the research exploring the factors influencing IR [72–74]. Among the identified works of literature, Bundhoo, Shah, and Surroop [75] used resilience theory as a framework for identifying the challenges of the resilience of energy infrastructure involved in restoring, rebuilding, and recovering energy security of Small Island Developing States (SIDS) to extreme weather conditions, such as cyclones, hurricanes, and floods; they discovered that the robustness of SIDS energy infrastructure and the long recovery time after suffering damage are two key factors causing the low resilience, and policy intervention is another potential factor affecting the resilience of SIDS energy infrastructure. Rasoulkhani and Mostafavi [76] proposed a multi-agent simulation model to assess the resilience and infrastructure dynamics that affect the long-term stability of civil infrastructure systems, and the proposed model captured three factors to shape the dynamics of the coupled human-infrastructure systems, including engineered physical infrastructure, human actors, and chronic and acute stressors; they pointed out that the performance mechanism of infrastructure systems is determined by their internal dynamics, and chronic stressors affect the effective performance of infrastructure systems. Simultaneously, these interacting factors work together to influence infrastructure resilience systems.

### 4.5. Prediction of Infrastructure Resilience

It is difficult to isolate causal, much less predictive, factors in resilience due to complicated and unknown disruptions [77]. In order to respond to disruptive factors in the resilience of infrastructure systems rapidly, some researchers devoted themselves to predicting infrastructure resilience via distinct means. For example, Mojtahedi, Newton, and Von Meding [78] adopted Cox's proportional hazards regression model to predict the resilience of transportation infrastructure to a natural disaster by determining the recovery rate and cumulative probability of recovery of the cross-regional transportation infrastructure in New South Wales, Australia [79,80]. Dhulipala and Flint [81] proposed a series of Semi-Markov process models to capture the inter-event dependencies in the recovery of infrastructure systems when successive disasters occur; it can realize the prediction of resilience, thereby affecting resilience-based decision-making.

## 5. Future Research Directions

Following a content analysis of identified papers on IR, the authors determined four future directions that may go towards in this field. The determination of these four research topics was primarily on the grounds of the expansion of major research streams and the feasibility of conducting new studies without weakening the significance of theoretical advancements.

### 5.1. Analyzing Factors Influencing Infrastructure Resilience Based on Simulation

According to the results of this review, it can be discovered that the majority of studies adopted traditional qualitative methods, such as literature reviews, to assess factors influencing infrastructure resilience, while few studies used simulation methods to analyze these factors from a dynamic perspective. Qualitative methods tend to rely on the empirical data from the researchers, whereas simulation is the tool capturing the operation of a real-world process or system over time. The organic combination of the two may generate a distinct spark for the field of IR. These factors influencing IR are usually classified into three categories, namely independent factors, correlated or cascading factors, and correlated or independent factors of the same nature. In addition to exploring the impacts of single independent factors on IR over time and space, simulation methods conduce to investigating the interaction of multiple correlated factors. Simulation-based analysis of impact factors contributes to showing their influencing process to IR intuitively, which can be considered a future research direction.

### 5.2. Assessing the Resilience of Green Infrastructure

The terminology "green" has been a topic of great concern in various fields in recent years. From the evidence of this review, there appears to be a strong focus on the correlation between green infrastructure and the improvement of urban resilience. However, literature is lacking in assessing the intrinsic resilience of green infrastructure. Green infrastructure is essentially a system that provides multi-purpose services to other infrastructures, including meeting the routine needs of the society and bringing additional health and well-being benefits [82]. The resilience of green infrastructure can affect the restoration of the entire ecological environment, which is also critical to the sustainable operation of urban systems [83]. What are the elements embody the resilience of green infrastructures? What are the factors influencing the resilience of green infrastructures? How do we measure and reinforce the resilience of green infrastructures? These issues need to be conducted in a further study in the future.

### 5.3. Improving the Resilience of Interdependent Infrastructure

The urbanization and the growth of the population are driving infrastructure systems in a complex and interconnected direction, which makes it critical to recognize the interdependence of infrastructure systems [84,85]. A complete infrastructure system consists of different categories of subsystems. If any of these subsystems fail due to natural disasters, social emergencies or other disruptions, urban areas will encounter higher hazards of cascading collapse [86]. In addition to investigating the resilience improvement strategies of individual infrastructures, it is also necessary to pay attention to the resilience improvement approaches of interdependent infrastructure systems. It seems that studies related to single infrastructure resilience are numerous, while the analysis of the effects of the interdependent infrastructure systems is relatively sparse. Exploring the improvement of the resilience of interdependent infrastructure can capture the actual requirements, which contributes to proposing effective strategies to enhance the capability of restoration after disruptive events.

### 5.4. Predicting the Resilience of Infrastructure Based on Empirical Research

When a disaster strikes, systems, services, and lifestyles in urban or rural areas will experience unprecedented shocks and pressure, causing many unpredictable and unavoidable events [87]. Thus, predicting the resilience of infrastructure systems responding to various disruptions is vital. According to the stream analysis in Section 4.5, there have been some studies about predicting infrastructure resilience in the past years, but most of them are concerned more with developing mathematic models or accurate algorithms to predict resilience, while studies that focus on applying the methods of prediction of resilience based on empirical research are not enough. To bridge this knowledge gap, future research should carry out empirical research applying various mathematic methods or structured models to predict the resilience of infrastructure systems in rural and urban areas, thus providing feasible strategies for enormous disasters.

## 6. Conclusions

In this study, a comprehensive literature review of IR studies published during 2011–2021 was conducted. This review identified and categorized relevant publications from various perspectives. It carried out an overview of the distribution analysis, including publications over time, publications by journal, publications by infrastructure type, top contributing researchers, contributing institutions, and contributing countries or regions. The review also provided the identification of research streams and potential directions to be implemented in the future.

In the distribution analysis, a growing interest in the research of IR is captured, particularly in the past five years; it is closely related to the fact that the issues related to sustainability and resilient cities or areas are highly valued by a variety of industries. These results also reflected that papers about IR were published in a wide range of journals, and

the scopes of these journals mainly focus on sustainability, safety, resilient infrastructure, and risk management; this aids in the selection of journals for submitting relevant papers. In addition, it can be discovered that half of all identified papers investigated the resilience issue in critical, general, and interdependent infrastructures, which shows the necessity of analyzing the resilience of a particular infrastructure in the future. Moreover, these results suggest that there has been a huge advantage in research fields of infrastructure resilience in major developed countries such as the United States and the United Kingdom due to their researchers' contributions. Meanwhile, institutions of these countries, such as the University of Florida, Arizona State University, and Texas A&M University, contributed a lot to infrastructure resilience research. The aforementioned results demonstrate that cooperation and connection among countries or regions, institutions, and researchers should be further strengthened to facilitate the deep exploration of IR issues.

In a detailed analysis, the review investigated major research streams involving the concepts, impact factors, assessment, improvement, and prediction of IR, and they also cover qualitative and quantitative analysis. However, the majority of publications focused on general infrastructure systems or utilized traditional research methods. Subsequently, four research directions for further research were determined in accordance with the weak research component, including analyzing factors influencing infrastructure resilience based on simulation, assessing the resilience in ecosystem and green infrastructure, improving the resilience of interdependent infrastructure, and predicting the resilience of infrastructure based on empirical research.

This paper provides a critical overview of infrastructure resilience in the academic field to acquire meaningful insights into the issue of infrastructure resilience. It also helps researchers quickly learn about existing research on infrastructure resilience, including research streams and research methods so that they can determine future research directions or research contents.

**Author Contributions:** W.L. wrote the original draft preparation and conducted data curation and formal analysis; M.S. and S.Z. were responsible for the review and supervision; X.Z. and Z.Z. were responsible for the conceptualization. All authors have read and agreed to the published version of the manuscript.

**Funding:** This research was funded by the <National Natural Science Foundation of China #1> under Grant [Number 71901224], <Natural Science Foundation of Hunan Province#2> under Grant [Number 2020JJ5779], and <Changsha Municipal Natural Science Foundation#3> under Grant [Number KQ2014116].

**Institutional Review Board Statement:** Not applicable.

**Informed Consent Statement:** Not applicable.

**Data Availability Statement:** Data sharing not applicable to this article.

**Conflicts of Interest:** The authors declare no conflict of interest.

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
