# Peer review of "Resilience in Infrastructure Systems: A Comprehensive Review"

_buildings, doi:10.3390/buildings12060759_

Round 1

Reviewer 1 Report

I think section 4 is the most important part of this review paper, but this section does not have adequate quality. Below are examples that authors can revise easily.  

  1. The subsection needs to be reordered based on Table 6. For example, 4.1. Assessing Infrastructure Resilience; 4.2. Improving Infrastructure Resilience; 4.3. Decomposing Infrastructure Resilience; 4.4. Predicting Infrastructure Resilience. 
  2. I could not find the importance of figure 4
  3. Subsection 4.1: what do you want to state? Is Table 7 needed? The title needs to be revised.
  4. Subsection 4.2: The title needs to be revised (e.g., factors influencing infrastructure resilience). This focused on the methodologies of previous studies for defining factors. I recommend major findings would also be introduced.
  5. Subsection 4.3: The title needs to be revised (e.g., assessment of infrastructure resilience) 
  6. Line 282: what is the five metrics?
  7. Line 291: some researchers? It needs to provide previous studies with citations.
  8. Line 309: A number of researchers? Same with Line 291
  9. Figure 6: please edit. Sizes of squares are different, and some lines are cut.
  10. Line 424: before Thus, it needs space.  
  11. Line 311: What is the multi-hazard assessment framework?
  12. Line 314: what is the year of occurrence? What methodologies were used to present the recovery curve?

Reviewer 2 Report

The submitted manuscript aims to bridge the knowledge gap in the field of the resilience of infrastructure systems by presenting a comprehensive review of infrastructure research conducted in the past decade, namely, from 2011 to 2021. On the basis of a systematic search, this study identified 222 journal articles investigating infrastructure resilience. A review of the identified paper revealed five research streams in the area of infrastructure resilience, namely, decomposing infrastructure resilience, influencing infrastructure resilience, assessing infrastructure resilience, improving infrastructure resilience, and predicting infrastructure resilience. The presentation of the paper is acceptable, the importance of the idea is appropriate, and I recommend the submitted manuscript should be accepted after regarding the following minor comments:

  1. The literature review has not been done for the conducted research in 2022.
  2. It is highly recommended to justify the research methodology based on the types of research that have been reviewed.
  3. How the authors have concluded those five research streams in the area of infrastructure resilience?
  4. Please clarify the notion/reasons of the logical line of illustrating the major research efforts in the considered field.
  5. It is highly recommended to double the check the presented articles in the field of green infrastructure.
  6. Figures should be located centred (for instance some parts of Figure 6 are not shown completely).
  7. Section 5 (Future Research Directions) should be elaborated.
  8. Has your paper proof read by a native English speaker or a person more familiar with the English language?

Round 2

Reviewer 1 Report

Citation: (author, year). For example, L 28: (Feofilovs and Romagnoli, 2017)

Did you follow PRISMA guidelines and complete the checklist and flow diagram?

Between subsections and sections, it needs to have transition sentences.

In Chapter 4, it is hard to see the need for future studies.  Between chapters 4 and 5, please add transitions. And it needs more explanation why you suggested those future research directions. 

L 19 – 20: please keep the consistency of the structure of the sentence. For example, you could change as improvement of infrastructure resilience, and prediction of infrastructure resilience.

L 30: and pumps are (no comma)

L 37: a lot from a variety of disasters – need to rewrite  

L 40: $300 billion – worldwide? Or in a specific country?

L 51: Does this sentence (The definition of resilience varies slightly depending on different fields) need?

L 61:  I think there are many review papers related to infrastructure resilience. You may need to specify the areas that show lacking. 

L105: no indent before “where”

L107: I am not sure about the importance of Table 1 and L 100 to L 107.

L126: Table 2 presents

L 158: Did you determine the scores in Tables 3, 4, and 5 using equation 1 in Line 104? Please make sure how you determined the scores.   

L 184 – L186: Please use the same structure of language and make sure the order. For example, “These research streams include assessment of infrastructure resilience, improvement of infrastructure resilience, the conceptualization of infrastructure resilience, factors influencing infrastructure resilience, prediction of infrastructure resilience.” With changes, please make sure the headings of Table 6 and titles of subsections.  

L 215 – L 218: Rewrite: For example, Toroghi and Thomas (2020) used a framework including five metrics (robustness, resourcefulness, redundancy, rapidity, and readjust-ability) which assesses the resilience of electric infrastructure systems using 4R concept and readjust-ability.

L224 – L 226: Please rewrite this sentence. For example, “various research assessed the resilience of infrastructure in terms of robustness (Bocchini et al. 2014, Sela et al. 2017, Homayounfar et al. 2018, Huizar, Lansey and Arnold 2018).” And when you add citations, please make sure the order (I presume it needs to order by published year).

L 226:

(1)  Why did not include this (Bollinger and Dijkema (2016)) in the list of citations, such as before Homayounfar et al., 2018? 

(2)  Is this study (Bollinger and Dijkema (2016)) related to robustness? 

L 228: what is the previous work? 

L 247: why did not you include this study (Cimellaro, Solari, and Bruneau (2014)) in the list of previous studies in Line 241?

L 247: recorded during the March 11 2011? 

(1)  Recorded only one day? 

(2)  March 11, 2011

(3)  Please rewrite this sentence

L 257: economical approach?  -> efficient approach?

From L 269 (Improving Infrastructure Resilience):

I think it would be better to organize previous studies in terms of the theme of studies.

For example, you can add the sentence, such as “there are previous studies for enhancing resilience using green infrastructure.” After that, you can introduce previous studies conducted by Lee et al., (2021) and Staddon et al., (2018). 

Similarly, you can add a transition sentence to begin a new paragraph and introduce the other works (Johansen and Tien (2018) and Zhao, Liu, and Zhuo (2017)).

L 314 to L 320: author and year do not match at the beginning and end of the paragraph.  2018 or 2019? Who is the first author? You don’t need to add citations at the beginning and end of the paragraph.

L 322: originally originates? 

L326 to 331: rewrite the sentence

Table 7: it requires the space between community/society and government.

L 359: please add the citation and reference for the Cox model.

Round 3

Reviewer 1 Report

Overall, the authors addressed my comments. However, I recommend a native speaker to read and revise this manuscript before publication.